# SpeedLoader: An I/O efficient scheme for heterogeneous and distributed LLM operation

**Yiqi Zhang**
Institute of Data Science
yiqi.zhang@u.nus.edu

**Yang You**
School of Computing
youy@comp.nus.edu.sg

National University of Singapore
Singapore, 119077

## Abstract

With the surging growth of model parameters, foundation models pose unprecedented challenges to traditional computational infrastructures. These large models inherently require substantial accelerator memory to accommodate massive tensors during pre-training, fine-tuning, and even inference stages, making it even more challenging to deploy a model with restricted computational resources. Given this challenge, distribution and offloading the model states are two major solutions. Partitioning the required states to participating workers, and storing them in lower speed media, such as host DRAM and block devices, largely alleviate the accelerator memory pressure. However, the prohibitive costs of tensor communication render it a theoretically plausible yet practically inefficient solution. Previous efforts to improve efficiency include maximizing rematerialization and employing chunk-based tensor management to reduce host-device communication. Despite these efforts, the reported training throughput only achieves 36.54% of model FLOPs utilization (MFUs), still not comparable to full on-device training. In this work, we redesign the data flow of heterogeneous hardware and sharded model training to minimize the excessive communication overhead. Our proposed scheme significantly enhances training and inference throughput of large language models under restrictive computational resources. We confirmed a large leap in effective compute time by looking into the kernel-level runtime behavior of our trials, where the MFUs can achieve up to 51%. Compared to the state-of-the-art approach, our framework robustly achieves remarkable speedups from 3x to 30x in multiple distributed heterogeneous training setups and inference speedups of 1.5x to 2.35x without compromising arithmetic precision.

## 1 Introduction

The trend of increasing transformer-based model sizes marks a paradigm shift in natural language processing deep neural networks. Research on model scales suggests that expanding transformer sizes significantly benefits aspects such as performance, universality, and transferability [1, 2]. In accordance with scaling laws[1], large language models (LLMs) have surged in recent years. For instance, models have grown from the 175 billion parameters in GPT-3 to over 400 billion parameters in LLaMA-3 [3, 4], the size of LLMs underwent a substantial leap. However, the continually growing size of these models places mounting pressure on computational infrastructures. For example, training

38th Conference on Neural Information Processing Systems (NeurIPS 2024).

GPT-3 175B requires approximately $3.14 \times 10^{23}$ floating point operations (flops[1]), equivalent to around 280,000 GPU hours on an NVIDIA A100 at its peak performance, which is not pragmatic without large scale parallelism.

The scalability of LLMs heavily depends on parallelism and distribution, with a significant constraint being the heavy memory footprint. Various techniques, such as tensor parallelism[5, 6], pipeline parallelism[7, 8, 9], and quantization[10, 11], can help alleviate this pressure. Model parallelism, in particular, involves sharding the model states (parameters, optimizer states, gradients) across accelerators to overcome limitations imposed by finite on-device high-bandwidth memories (HBMs) [12]. Unfortunately, HBMs remain scarce and costly, imposing significant infrastructural constraints on training and using larger models. For example, a half-precision 1.8 trillion parameter model demands over 180 units of 80GB GPUs to hold the model and optimizer states, not accounting for the memory needed for essential activations. Relying solely on HBMs for state storage is becoming impractical with current accelerator technology as models continue to scale.

Offloading has emerged as a flexible solution for both training and inference with LLMs across various infrastructure scales. It allows model states to be stored on larger media like host DRAMs and NVMe drives, loading them to the accelerator only when required for computation [13, 14, 15]. However, the merits of offloading come at the expense of even higher I/O overhead. Model weights must frequently shuttle between accelerator and host memory during computation. Additionally, during inference, large volumes of key value caches (KV Cache) [16], need to be moved to external storage and subsequently brought back to the GPU for the next token. The overhead caused by data movement often surpasses computation time due to bandwidth bottlenecks between accelerator, CPU, and NVMe storage, resulting in high task latency.

To minimize redundant communication and maximize hardware utilization, we propose SpeedLoader, an highly optimized scheme to eliminate I/O redundancies through computation rescheduling during LLM operation with model sharding and offloading. By meticulously feeding and offloading activations, SpeedLoader computes multiple batches with only two full-model loading. We evaluated SpeedLoader's performance with LLaMA-2 and OPT [17, 18] at different sizes. Results showed that SpeedLoader can robustly achieve a training speedup of 3.5x to 30x and over 50% model FLOPs utilization (MFU) on multiple platforms compared to state-of-the-art approaches.

The major contributions of this work are: (1) We proposed a compute strategy that minimizes peer and device-host I/O in heterogeneous and sharded LLM training; (2) We implemented a high-efficiency tensor exchange manager that transparently shuttles tensors between device and host, minimizing fragmentation and redundancy; (3) Our optimized tensor management enables higher inference efficiency than previously state-of-the-art (SOTA) approaches.

## 2   Related Work

LLMs are resource-intensive and typically require specialized strategies to alleviate hardware stress. One common approach is to optimize distribution and parallelism across multi-GPU clusters. For instance, Megatron-LM [5] leverages advanced model parallelism techniques to achieve scalability, and Zero Redundancy Optimizer (ZeRO) partitions and distributes all model states (i.e., parameters, gradients, optimizer states) among GPUs, recollecting them only when the layer needs to be computed [12]. Similarly, Fully Sharded Data Parallelism (FSDP) shards both model parameters and input data across multiple GPUs[19]. These techniques usually gather full parameters only when needed and reduce-scatter the newly computed gradients across peers, significantly alleviating memory pressure on the accelerators while introducing considerable communication overhead.

While distribution frameworks still require substantial hardware, offloading approaches address memory shortages in a different way. Work including ZeRO-Offload, Capuchin, and SuperNeurons [13, 20, 21] enables large model training by offloading data and computation to host RAM, mitigating the memory and computational demands of training large models with fewer GPUs. ZeRO-Infinity

---
[1]A list of abbreviations in this paper can be found in Tab. 3.

[14] extends this by offloading more data to high-speed block devices like NVMes, overlapping computation and communication for higher bandwidth utilization. During inference, offloading can significantly reduce the computing resources required for pre-trained models of unprecedented scale. Unlike early works, PatrickStar[22] organizes the model data in memory chunks and dynamically distributes them in the heterogeneous memory. FlexGen [15] offers a wider range of batch size choices with an efficient offloading strategy, and thus significantly increases maximum throughput, demonstrating that high-throughput generative inference of a 175 billion parameter LLM is feasible with a single GPU. Current offloading solutions [13, 22] support prefetching parameters after the initial forward/backward pass, and the granularity of overlapping can be precisely controlled[23]. As an update, ZeRO++ implements a hierarchical weight partition, which maintains a full model copy on each node to largely reduce inter-node tensor exchange [24].

In addition to these two primary solutions, other approaches have been explored. Rematerialization techniques [25, 26, 27] recomputes activations on-the-fly during backpropagation rather than storing all intermediate activations in memory. Model compression methods such as quantization are also commonly used to reduce memory requirements[10, 11]. However, compression usually comes at the cost of numerical inaccuracy and retraining of the model.

# 3    SpeedLoader

## 3.1    Overview

In canonical solutions (Fig. 1, left), each mini-batch is processed sequentially, requiring the full model to be loaded twice per batch. In distributed settings, the accelerator must read local partitions of model states from host DRAM and gather other partitions from peer ranks. Additionally, to properly partition the accumulated gradients, reduce-scatter operations across all ranks are inevitable, introducing communication equivalent to one copy of the model weight during each backward propagation.

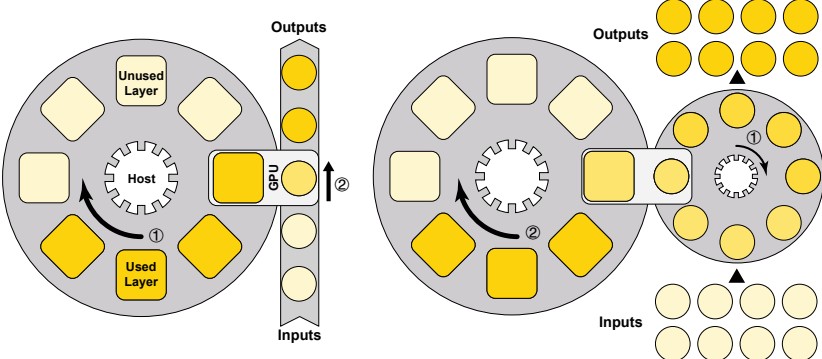

Figure 1: Mechanism comparison between canonical approach and SpeedLoader. **Left, canonical approach** ① load the whole model every time ② loads a single batch; **Right, SpeedLoader** feed every layer with multiple batches and exchange activations to and from host memory (①). Therefore, for one full model loading(②), SpeedLoader can process multiple batches.

Here we propose SpeedLoader, which processes multiple sub-batches, collectively referred to as an effective batch (i.e., number of sub-batches), with only two model loading and gradient synchronization (Fig 1, right). For any incoming batches shaped $(N, l)$, they are evenly split to sub-batches along batch size axis $(n_s b, n, l)$, where $N, n_s b, n$, and $l$ are effective batch size, number of sub-batches, sub-batch size and sequence length, respectively. Each time a layer is loaded, new inputs are fed into the model while the activations from the previous sub-batch are offloaded to a host buffer to prevent memory spikes. After processing all sub-batches with the current layer, the accelerator releases the current layer and proceeds to the next layer. Subsequently, the previously saved activations are reloaded. This method achieves gradient accumulation with an altered computation graph, with

significantly reduced frequency of model state communication. A quantitative analysis is discussed in Sec. A.3.

The backward propagation process also needs to be redesigned. By default, each sub-batch generates an individual loss, and the automatic differentiation package conducts backward propagation on each sub-batch sequentially. This process loads the model and partitions gradients once per sub-batch, leaving the prefetching of associated gradients and activations out of our control.To address this issue, SpeedLoader redesigned the computation graph. We re-route the computation graph by connecting activations from the same layer with a no-op function. This modification allows backward propagation to trace back in the same manner as forward propagation. Additionally, the connecting function buffers and passes correct gradients when switching activations, ensuring scheme correctness. In this way, each module computes and accumulates the gradient with respect to input and loss of every individual sub-batch. This approach enables the automatic differentiation mechanism to function correctly without causing excessive I/O.

### 3.2    Tensor Exchange Manager

The core mechanism of SpeedLoader lies in novel scheduling of tensor computation and exchange, particularly in managing and minimizing communication overheads. Heavy data exchanges can lead to memory fragmentation, resulting in memory waste and reduced hardware efficiency. To address this, we carefully examine the details and propose a simple yet effective technique to optimize performance.

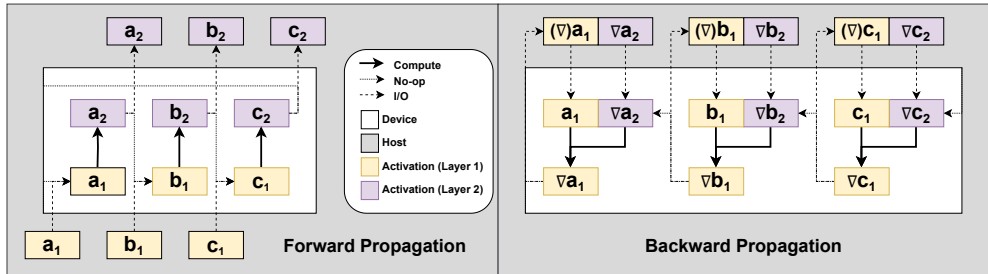

Figure 2: Device-host communication breakdown of SpeedLoader.

**Transparent data exchange.** The implementation of SpeedLoader provides a plug-n-play function wrapper. Users can wrap the forward function of each layer in the model to iteratively load and compute multiple sub-batches in one explicit forward call. Taking a closer look at the wrapper demonstrated in Fig. 2 and Algorithm 1, it performs the following activities before and after the forward/backward calls:

Here, pinned_x is a pre-allocated host memory for activation and gradient storage. Before each forward pass, the activation of previous sub-batch is offloaded while prefetching the input of the next sub-batch simultaneously. This communication is also overlapped with the computation of current sub-batch using two separate CUDA streams in our implementation. During each backward propagation, the gradient of the previous sub-batch activation is offloaded, and the next required activation and gradient are prefetched. Similar to the forward propagation, these two processes are overlapped with the re-computation of checkpointing and computation of gradients. In inference scenarios, the communication is much simpler: assuming KV caching is enabled, we exchange the KV cache instead of the entire activation.

**Fragmentation-free Memory Pool.** Without prior information of the model, PyTorch can only adopt an on-demand approach to allocate host memory for offloaded tensors[28]. However, allocating extensive continuous memory in this manner is a lengthy and blocking operation. Meanwhile, excessive memory allocations can result in significant memory fragmentation, and even highly optimized LLM serving systems can suffer from heavy fragmentation[29]. Given the configuration of a transformer-based LLM, it is possible to pre-allocate a reusable memory pool. In this study, we

allocate a minimal page-locked memory pool for caching offloaded activations or KV caches based on the given hyperparameters of the model.

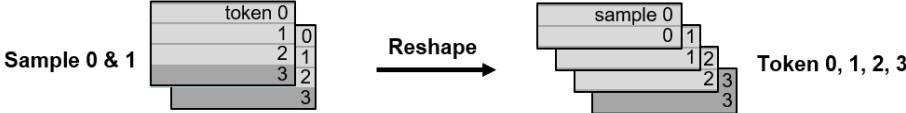

Figure 3: Reorganizing buffer memory to ensure continuity.

**Algorithm 1** Pseudo-code for tensor exchange.

$x \leftarrow \text{embedding}(input\_ids)$
**for** $i = 1$ **to** len(batches) **do**
    Offload $x$ to pinned_x$[lid][i-1]$
    $x \leftarrow$ buffer
    Register_hook($x$)
    Fetch pinned_x$[lid-1][i+1]$ to buffer
    $x \leftarrow \text{layer}(x)$
**end for**
$output\_logits \leftarrow \text{output\_embeddings}(x)$

**procedure** BACKWARD_HOOK($x, i, lid$)
    Offload $x$.grad to pinned_x$[lid-1][i-1]$
    $x \leftarrow$ act_buffer
    $x$.grad $\leftarrow$ grad_buffer
    Fetch pinned_x$[lid-1][x-1]$ to act_buffer
    Fetch pinned_x$[lid][x-1]$ to grad_buffer
    backward($x$)
**end procedure**

Additionally, this pinned memory pool is highly optimized for sparse KV caching under inference workloads. KV cache within the same sub-batch is preserved for each layer throughout the generative steps to maintain computational consistency. In our approach, only the KV state of the new token is written to the host memory. To ensure the continuity of the host pinned memory buffer, the buffer pool is organized in the format of ($n\_sub\_batches$, $layer$, $sequence\_length$, $-1$). Given the augmenting nature of sequence length through generation steps, it is permuted to the third dimension from its original position in the sixth dimension (Fig.3). Therefore, a contiguous memory chunk is available for asynchronous bulk copy for all tokens in the sub-batch, eliminating the need for blocking operations or additional loops.

### 3.3 Hyperparameter Tuning Strategies and Communication Analysis

Hyperparameter selection is a critical aspect of both the training and inference phases of LLMs. Our proposed method expands the search space for hyperparameter tuning, highlighting the importance of a swift tuning strategy. Based on observations in Section 5.1, we have developed a one-shot hyperparameter tuning strategy that not only addresses these new dimensions (i.e., sub-batch size, effective batch size and number of on-device layers) but is also compatible with existing framework-provided tuning tools. Details of those hyperparameters can be found in appendix A.2. Our observations show that the performance of SpeedLoader significantly benefits from the exhaustive usage of available memory, including accelerator HBM and host DRAM.

To achieve optimal training and inference throughput, we aim to minimize data movement and maximizing arithmetic density iterating through the same amount of training data. In this context, we will discuss scenarios with the assumption that gradient checkpointing is enabled for training. Given $P$, $A$, $N$ refering to parameter, activation size and number of sub-batches, SpeedLoader has $5NA + 3P$ and $3P$ local and remote communication, compared to both $3NP$ for ZeRO-Offload method (see quantitative details in appendix A.3). As a result, the I/O advantage of SpeedLoader over unmodified ZeRO-Offload can be asymptotically enhanced by increasing effective batch size of SpeedLoader.

The one-shot hyperparameter tuning primarily aims to maximize the utilization of HBMs and host memory. Given a target context length, the tuner runs two iterations of the training script with different batch sizes, recording the maximum allocated host and device memory during these runs. Since memory consumption is highly linear with respect to the number of input tokens (as discussed in Section 5.1), the tuner calculates the memory increment per input entry. Using this information, it

determines the maximum sub-batch size and the number of sub-batches that can be accommodated within the available device and host memory. This approach allows for efficient utilization of memory resources, ensuring optimal performance.

# 4    Evaluation

**Implementation.** For any transformer-family model, we provide a wrapper function that iteratively feeds the model with prefetched activations. This approach allows the model to compute and offload activations of several sub-batches without exceeding device memory limits. The implementation also includes utilities that automatically tune hyperparameters. Our current implementations support both OPT and LLaMA-2 [17, 18]. Other transformer-based models can be integrated with minimal effort. For stringent comparison, we benchmarked the LLaMA-2-7B, 13B, and 70B models. To establish fair comparison with FlexGen[15], we also benchmarked OPT-6.7B, 30B, and 175B. For optimal offloading and scalability, SpeedLoader is implemented based on DeepSpeed ZeRO++ with hierarchical weight partitioning[24]. Additionally, 16-bit brain floating-point and half precisions are employed to minimize memory pressure and maximally utilize tensor cores on NVIDIA GPUs.

**Platform specifications.** Our experiments were performed on VMs from an Infrastructure as a Service (IaaS) provider and computing nodes from a high-performance cluster (HPC). Our benchmarks were conducted on VMs with 16 NVIDIA A100-40GB GPUs with NVLink. Each VM is equipped with 96 cores of vCPU and 1360GB RAM. Functionality tests were conducted on HPC nodes with NVIDIA A100-40GB GPUs and HPE Slingshot Interconnection in Dragonfly topology. Platform specifications can be found in Tab. 1.

Table 1: Platform specifications for each experiment

| Experiments | Internode Connection | Intranode Connection | Accelerator |
|---|---|---|---|
| Functionality | | HPE Slingshot | |
| Benchmark | NVLink,PCIe Gen 4 | 100Gbps Ethernet | NVIDIA A100-40GB |
| Profiling | | 100Gbps Ethernet | |
| Scalability | | 100Gbps Ethernet | |
| | PCIe Gen 5 | | NVIDIA H100-96GB |
| Compatibility | PCIe Gen 3 | N/A | NVIDIA V100S-32GB |
| | PCIe Gen 4 | | NVIDIA A6000 |

# 5    Results

## 5.1    Impacts of Hyperparameters

We first analyzed the impact of changing hyperparameters (Fig.4) with a model with LLaMA-2-7B on single NVIDIA A100 GPU.

**Linear resource usage.** Our analysis indicates that both HBM and host DRAM usage exhibit a linear pattern influenced by major hyperparameters. Using ordinary least squares regression, we found a highly linear relationship ($R^2 > 0.999$) between sub-batch size/number of on-device layers and peak GPU memory usage, as well as total tokens and host residence memory usage, as shown in Fig. 4. Intriguingly, the tensor exchange memory pool allocates more page-locked memory than expected. This is rooted in the memory allocation behavior of PyTorch CPU tensors, which automatically rounds up the reserved memory to the nearest power of 2. Despite this linearity and predictable host memory allocation, predicting memory consumption on the accelerator remains challenging and can vary significantly across different models. To maximize the generalizability of our proposed approach, we decided to adaptively fit the linear resource model on the fly.

**Asymptotic compute efficiency.** We identified that the MFU is strongly affected by the hyperparameters including sub-batch size, number of sub-batches, and number of on-device layers. Increasing

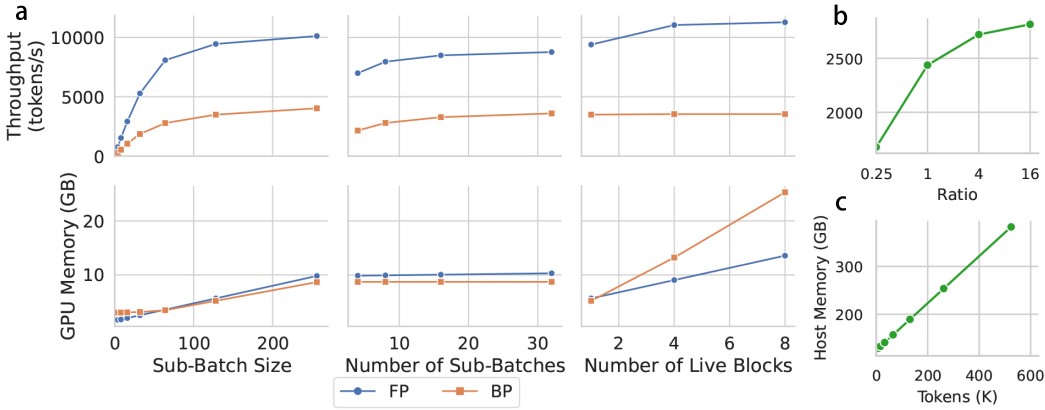

Figure 4: Hyperparameter Analysis. **a,** the dynamics between throughput, GPU memory consumption and hyperparameters, where FP and BP denotes forward and backward propagation, respectively; **Ratio** in panel **b** refers to the ratio between sub-batch size and number of sub-batches; **c,** total tokens' effect on host memory usage.

each hyperparameter can result in enhanced arithmetic intensity during computation. Aligned with theoretical analysis, the increasing hyperparameters asymptotically saturates the computation to the practical peak performance of the model operations. Considering the host DRAM as a bottleneck, the maximum token capacity of a given system is fixed. Given this restriction, the tradeoff between sub-batch size and the number of sub-batches becomes critical. Both hyperparameters are positively correlated with throughput, however, they have different degrees of impact. We conducted a brief test to examine the effects in action. The results (see Fig. 4b) show clearly that the model achieves optimal performance when the sub-batch size is maximized to the limit allowed by HBM.

**Performance degradation on long sequence.** With the total token number saturating the HBM, we adjusted the sequence length and sub-batch size ratio to test the trade-off between these two HBM-bounded properties. The results show a decrease in total throughput with growing sequence length and shrinking sub-batch size. However, the sequence length is usually determined by the data and is not a tunable hyperparameter. This observation does not alter the objective of filling all available HBM during hyperparameter selection.

## 5.2 Enhanced Arithmetic Intensity

Detailed profiling results further demonstrate that our approach substantially enhances the utilization of the given hardware. By properly applying our scheme, the effective GPU kernel execution ratio rises from 8.9% to 83.7%, as shown in Fig.5. The baseline approach spent nearly 80.6% of the time exchanging tensors to and from host DRAM, while SpeedLoader controls the memory accessing ratio to under 15%.

For distributed training with ZeRO stage 3 without offloading, the baseline approach spent 80% of the time partitioning the gradients and 14% of the time gathering weights. In contrast, SpeedLoader achieves 66% GPU time executing arithmetic operations.The model operation exhibits distinct behaviors between saturated and unsaturated computations. When the sub-batch size is large enough, kernel executions are seamlessly enqueued in series, typically lasting longer than the host-side function calls of the active layers. Conversely, when the sub-batch size is not large enough to fill the CUDA stream, the accelerator often waits for host-side function calls. The backward computation follows a similar trend (Fig. 5b,c).

Additionally, DeepSpeed backward computation incurs a significant blocking communication load, which offloads the gradients to pageable host memory. For every checkpointed segments of our model in trial, this communication operation causes 2110 ms of hardware turnaround without any arithmetic

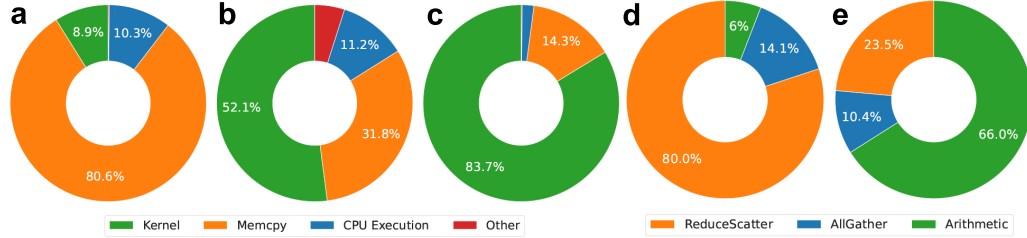

Figure 5: Profiling results. (**a**) Deepspeed (w/ offloading); (**b**) SpeedLoader (unsaturated-computing, w/ offloading); (**c**) SpeedLoader (saturated-computing,w/ offloading); (**d**) Deepspeed (w/o offloading); (**e**) SpeedLoader (w/o offloading)

operations. On the other hand, the corresponding backward propagation arithmetic operations take 6679 ms, indicating that nearly a quarter of the backward propagation time is spent waiting for blocking gradient partitioning. In conclusion, the sub-batch size saturates computation, while the number of sub-batches hinders the synchronization overhead in the computation of every effective batch.

Another interesting observation is that the prefetching operation of DeepSpeed is module-wise asynchronous. Instead of overlapping the communication with all operations before the target tensor is called, only one module's operation is allowed to be overlapped with a single I/O operation. This behavior results in a bottleneck at the end of every active layer group, where parameters of current layer group are released and prefetching kicks off.

## 5.3 Peak Performance Benchmarking

To demonstrate the pragmatic performance enhancements, we examined the peak performance of different offloading and distribution schemes on the same nodes from an IaaS provider. The results are shown in Fig. 6.

The first benchmark was conducted on a single device to evaluate SpeedLoader's offloading performance. Under the best configuration for both approaches, our method showed speedups of 3.29 and 6.49 for LLaMA-2-7B and LLaMA-2-13B, respectively. In both cases, SpeedLoader exhibited a MFU of 51%. The second test examined performance in ZeRO distributed training combined with offloading. We distributed the batches across 64 GPUs on 8 nodes interconnected with Ethernet. Our approach attained speedups of 5.48, 5.49, and 30.33 for the 7B, 13B, and 70B LLaMA-2 models, respectively. Notably, there was a MFU drop compared to the single device case, likely reflecting inter-node bandwidth bottleneck, which can be significantly alleviated in clusters with more reliable interconnections. The third test examined SpeedLoader's effectiveness on peer communication. In this trial, we disabled offloading compared to the previous test. Our approach increased the performance gap between the baseline, with speedups of 5.34 and 12.28 on the 7B and 13B models, respectively. This result highlights the potential of current sharded training paradigms, where the MFU can be elevated to unprecedented levels.

Additionally, our optimized inference scheme outperforms previous state-of-the-art approaches (Fig. 7). On a single NVIDIA A100, the speedups compared to FlexGen were 1.52, 2.09, and 2.35 for the OPT-6.7B, OPT-30B, and OPT-175B models, respectively. vllm showed consistently inferior performance due to its simplistic handling of offloaded computing. Note that the OPT-175B was tested with NVMe offloading and without KV caching due to excessive DRAM consumption. For OPT-175B, the KV cache for each sample with 256 tokens can consume up to 1.125GB of host memory, resulting in heavy I/O overhead and becoming a significant bottleneck in maximizing effective batch size. In contrast, our strategy without KV caching benefits from a larger sub-batch size and gains enhanced arithmetic intensity.

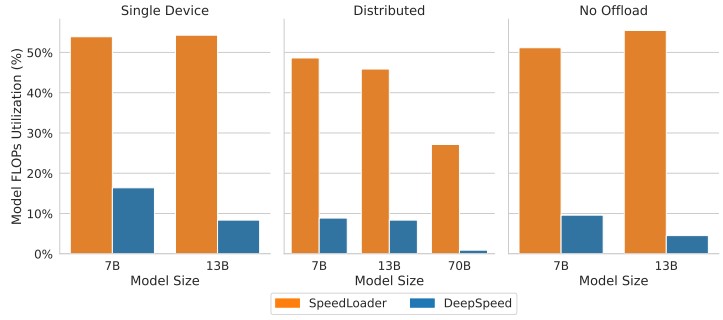
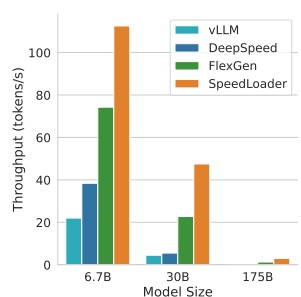

Figure 6: Training MFU comparison. **Left,** Single device benchmarking w/ offloading; **middle,** Distributed benchmarking w/ offloading; **right,** Distributed benchmarking w/o offloading.

Figure 7: Inference throughput benchmarking on single device.

## 5.4 Functionality Test and Scalability

Our methodology does not alter the model structure and numeric value. To show the performance of our proposed approach in real-world scenario, we also examine the functionality of our approach via pretraining experiment. We pretrained a 7B and a 13B model following the corresponding configuration of LLaMA-2. The trials ran on Wikipedia, OpenWebText and C4 datasets for a cutoff time. This cutoff was set before the full dataset was processed, establishing that training loss is a valid metric for assessing convergence effectively.

The training results are shown in Tab. 2. These trials were conducted using 4x NVIDIA A100 GPUs, distributed across four nodes interconnected by Slingshot Interconnection. They operated with an effective batch size of 512 and a context length of 2048. Given the cutoff time, our approach robustly processes more tokens than baseline implementation in 6 trials with three datasets and two model sizes without compromising convergence. The average speedup for 7B and 13B model are 4.08 and 2.68 respectively.

Table 2: Pretraining results of DeepSpeed and SpeedLoader

| Dataset | Model Size | DeepSpeed | | Ours | |
|---|---|---|---|---|---|
| | | Loss | Tokens (M) | Loss | Tokens (M) |
| Wikipedia | 7B | 2.896 | 26.2 | **2.507** | 113.2 |
| | 13B | 5.112 | 14.7 | **2.461** | 53.5 |
| OpenWebText | 7B | 4.409 | 27.3 | **3.705** | 116.4 |
| | 13B | 5.008 | 18.9 | **4.813** | 34.6 |
| C4 | 7B | 2.517 | 35.7 | **2.199** | 131.1 |
| | 13B | 3.544 | 18.9 | **2.058** | 48.8 |

The stark difference in convergence efficiency highlight the practical advantages of our proposed approach, especially when considering I/O operations. In this case, the standard approach required 32 gradient accumulation steps for a single parameter update, involving the model being loaded 64 times from the host. Additionally, to synchronize gradients and parameters, the amount of reduce-scatter and all-gather communications was equivalent to over 32 times the model size. Instead, our approach achieves same amount of computing with only two instances of parameter loading, together with only two full model all-gather and one reduce-scatter.

Our proposed scheme delivers excellent performance in distributed settings, especially for model sharding strategies like FSDP and ZeRO, SpeedLoader significantly boosts performance by largely reducing the parameter gathering cost with or without offloading. To examine the scalability of our approach, we conducted tests on 64 NVIDIA A100 GPUs on aforementioned cloud platform. The results are shown in Fig. 8.

In the weak scaling test, we scaled up the total number of accelerators with fixed workload per device with offloading enabled. We observed a superliner scaling pattern, where the throughput per device increased with the number of computing device. For LLaMA-2-70B, per device efficiency has over 3-fold speedup. This behavior is expected due to the bandwidth-centric design of ZeRO-Infinity[14]. With the engagement of more accelerators, the PCIe bandwidth bottleneck is largely alleviated. Additionally, our scheme can further reduce the inter-device and inter-node communication overhead for gathering weights and scattering gradients.

## 6  Limitations and Insights

SpeedLoader provides distributed deep learning settings with an I/O efficient option. By fully utilizing the available host memory, SpeedLoader can significantly reduce the redundant model state exchange among the device, peer and host memory. However, this approach may make host memory a new bottleneck for efficient model operation. Additionally, the implementation of memory allocation in PyTorch only allows sizes that are powers of two, which can lead to inefficient use of memory. For example, if a tensor requires 33 GB, PyTorch allocates 64 GB, potentially leading to out-of-memory issues. Currently, there is no explicit option to modify this behaviour for CPU memory allocation.

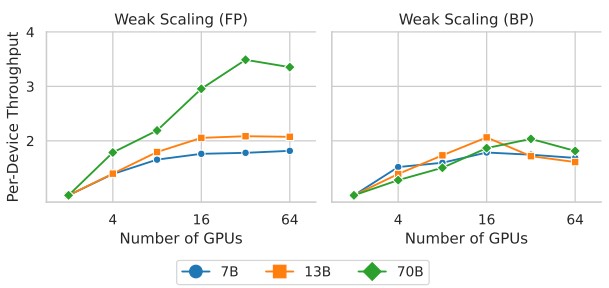

Figure 8: Weak scaling of SpeedLoader. Showing per device throughput.

Furthermore, while SpeedLoader is theoretically compatible with other parallelism strategies like tensor parallelism and pipeline parallelism, it has not yet been tested in conjunction with these methods. The potential benefits of SpeedLoader in larger-scale hybrid parallel settings remain to be explored. While data parallelisms like FSDP and ZeRO have low communication overhead in exchanging activations, tensor parallelism has lower communication overhead in exchanging parameters. Thus, exploring the dynamic trade-off with SpeedLoader between these parallelisms is a valuable area for future research. Amidst an age of LLM blooming, we hope that, together with other heterogeneous techniques, SpeedLoader can provide researchers with a new option to explore the frontiers of heterogeneous and distributed computing.

## 7  Conclusion

SpeedLoader is a highly optimized computing scheme for I/O-bounded distributed and heterogeneous LLM operations. By virtue of the layered nature of transformer models, our proposed approach retains layers *in-situ* to process more batches. Through meticulously designed tensor exchange, SpeedLoader enables efficient handling of multiple batches within a single forward-backward pass, significantly reducing the amount of required I/O. This not only allows users to maintain computational accuracy but also ensures time and energy efficiency by minimizing communication overhead. Consequently, SpeedLoader provides a practical, high-performance solution for I/O-constrained distributed and heterogeneous setups, where efficient large-scale model training and inference are key. Our results showed that SpeedLoader exhibits substantially higher performance than previously SOTA approaches during both the training and inference scenarios.

## 8  Acknowledgements

A special shout out to Dr. Huang Qirui, Dr. Wei Yuying, Ms. Nie Ying, and Dr. Wang Qinyi for their invaluable insights into this project and their selfless assistance in the writing of this work. The computational work for this article was partially performed using resources from the National Supercomputing Centre, Singapore (https://www.nscc.sg).

Yang You's research group is being sponsored by NUS startup grant (Presidential Young Professorship), Singapore MOE Tier-1 grant, ByteDance grant, ARCTIC grant, SMI grant (WBS number: A-8001104-00-00), Alibaba grant, and Google grant for TPU usage.

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

# A Appendix / supplemental material

## A.1 List of Abbreviations

Table 3: List of Abbreviations

| Abbreviation | Definition |
|---|---|
| flops | floating-point operations |
| FLOPs | floating-point operations per second |
| MFU | Model FLOPs Utilization |
| LLM | Large Language Model |
| HBM | High-Bandwidth Memory |
| ZeRO | Zero Redundancy Optimizer |
| FSDP | Fully Sharded Data Parallel |
| FP | Forward Propagation |
| BP | Backward Propagation |
| GEMM | General Matrix-Matrix Multiplication |

## A.2 Hyperparameters

Our approach introduces a few new dimensions of hyperparameters:

**Sub-batch size.** Due to the re-batching operation, the original batched data is parceled into smaller chunks, with the sub-batch size referring to the size of individual chunks. The aligned sub-batch size is a key hyperparameter that impacts computation. The major arithmetic pressure of transformers lies in linear layers, where batched general matrix-matrix multiplication (GEMM) is heavily used. Performance can greatly benefit from larger aligned batch sizes[30]. However, the total available on-accelerator HBM is the limiting factor for sub-batch size. The HBM usage is mainly shared by activation tensors, parameters in use, and exchange buffers, which is highly sensitive to other hyperparameters such as the prefetch range. This indicates that tuning of sub-batch size must take its interaction with other hyperparameters into account.

**Number of sub-batches.** As discussed in Section A.3, a larger effective batch size can result in asymptotic I/O reduction. A lower communication-compute ratio allows easier overlapping for optimal efficiency. However, offloading activations inevitably leads to significant page-locked memory consumption. Since ZeRO-Offload also applies enormous pressure to host memory, the host DRAM space becomes a major limiting factor for effective batch size.

**Number of on-device layers.** This hyperparameter specifies the number of model layers computed together. Instead of loading the model layer by layer, we compute multiple layers in one iteration. A larger number will result in more HBM consumption for buffering and storing active model parameters. Although this hyperparameter does not significantly affect model training or inference efficiency, it is essential for balancing host-device memory costs.

From our observations, we showed that the resource usage of our method has a linear relationship to the hyperparameters. Therefore, we can estimate the best batch configurations within only 2 trials by confirming the coefficient and intercepts.

## A.3 Communication Analysis

Taking a closer look at the communication, we can quantitatively analyze the communication amount in a single forward propagation and backward propagation. As listed in Tab. 4: In canonical approach, for every batch with n tokens, the model must be loaded twice from host for forward pass and re-computation, and one copy of gradient will be written back to host. In distributed settings, the parameters and gradients are communicated with peer ranks. To sum it up, the amount of local and peer tensor communications is both $3NP$ for every $N$ batches.

We illustrate the data operations of SpeedLoader in Fig 2. Take sub-batch b as example. During forward propagation, for each sub-batch at one layer, we prefetch the input of next mini-batch ($c_1$) and

offload the output of previous mini-batch ($a_2$). Meanwhile, we also prefetch the model parameters of the next layer. This sums to $2NA + P$ local and $P$ peer communication.

For backward pass with gradient checkpointing, the input activation ($a_1$) and the output gradient ($\nabla a_2$) are fetched. Meanwhile, previous activation gradient ($\nabla c_1$) is offloaded to the pinned memory that previously storing the activation ($c_1$). Simultaneously, the model parameter of next layer is also being fetched. Above results in $3NA + 2P$ local and $2P$ remote communication. In total, our proposed method has 5NA+3P and 3P local and remote communication, respectively.

Consider that a typical transformer model having a total size of $12Lh^2$ [31, 1], and checkpointed activation sizes are $nh$, the local and remote I/O ratios between SpeedLoader and ZeRO-Offload are $\frac{5n}{36Lh} + \frac{1}{N}$ and $\frac{1}{N}$, respectively. Based on the quantitative analysis, it is shown that the total communication during offloading in SpeedLoader is asymptotically reduced with increasing the number of sub batches.

Table 4: Communication during training, where $P$, $A$, $N$ refer to parameter, activation size and number of sub-batches respectively.

| Collective Communication | Ours | ZeRO |
| --- | --- | --- |
| Parameter Gathering (FP) | $P$ | $NP$ |
| Parameter Gathering (BP) | $P$ | $NP$ |
| Gradient Reduce-Scatter | $P$ | $NP$ |
| **Total** | $3P$ | $3NP$ |

| Local Communication | Ours | ZeRO |
| --- | --- | --- |
| Parameter Loading(FP) | $P$ | $NP$ |
| Activation Loading(FP) | $NA$ | - |
| Activation Offloading (FP) | $NA$ | - |
| Parameter Loading (BP) | $P$ | $NP$ |
| Parameter Gradient Offloading | $P$ | $NP$ |
| Activation Gradient Loading | $NA$ | - |
| Activation Gradient Offloading | $NA$ | - |
| Activation Loading (BP) | $NA$ | - |
| **Total** | $5NA + 3P$ | $3NP$ |

## A.4 Enhanced Performance with FlashAttention-2

We explored the synergy between SpeedLoader and FlashAttention-2[32]. We conducted experiments under identical hyperparameters with FlashAttention-2 enabled and disabled, ensuring maximum hardware utilization. Results (Fig. 10) showed that in distributed offloading scenarios, FlashAttention-2 provides 1.13x, 1.13x, and 1.05x speedups for LLaMA-7B, -13B, and -70B, respectively. In distributed training without offloading, the speedups are 1.16x and 1.13x for 7B and 13B models. For these tests, we used 1, 2, and 4 nodes with 16 NVIDIA A100-40GB GPUs each for 7B, 13B, and 70B trials, respectively. With this optimization and the upcoming FlashAttention-3, SpeedLoader can achieve throughput very close to non-sharded and non-offloaded training.

## A.5 Compatibility Test

SpeedLoader is designed to facilitate computing in I/O bounded settings. Therefore, link speeds of benchmark platform have substantial impacts on the performance of our solution. To explore SpeedLoader's performance on devices with various I/O capabilities and architecture, we tested the speedups of SpeedLoader on multiple distinct platforms. Details of benchmark platforms can be found in Tab. 1.

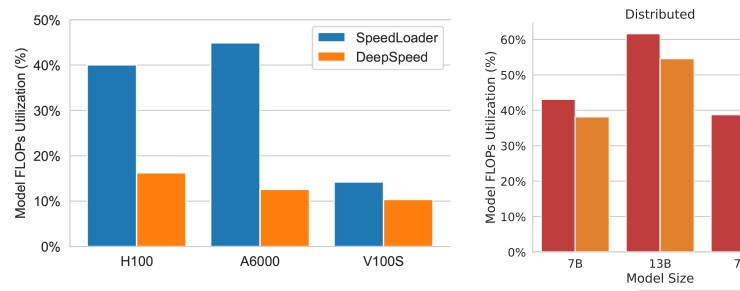

Figure 9: Single device peak performance test on devices with different bus speed.

Figure 10: Training MFU ablation study with FlashAttention-2.

With speedups of 1.37x, 3.56x, and 3.01x on V100S, A6000, and H100 (Fig. 9), SpeedLoader exhibited consistent gains on platforms with PCIe Gen 4 and over, and lower speedups on V100S with PCIe Gen 3. Overall, the positive speedups showcase a robust compatibility on platforms with mixed architectures and capabilities. Note that the MFU being lower on H100 than A100 is expected behaviour. As reported by Dao et al.[33], without architecture-specific optimizations like FlashAttention-3, transformers struggle to unleash the full potential of Hopper GPUs.

## A.6 Reproducibility Statement

To facilitate reproduction of experimental results, we provide an public repository[2]. As an example setting, we recommend readers reproduce part of aforementioned results with instances of A2 VMs on Google Cloud. This SKU has similar specifications to the computational resources of this research.

---

[2]Access here

