# OpenReview forum: "SpeedLoader: An I/O efficient scheme for heterogeneous and distributed LLM operation"
_NeurIPS.cc/2024/Conference — NeurIPS 2024 poster_

### Official Review · Reviewer_Dtr1 · 2024-07-11

**Soundness:** 2
**Presentation:** 1
**Contribution:** 2
**Rating:** 4
**Confidence:** 4

**Summary:**

This paper propose a new scheduling mechanism via effective sub-batch centric computation, to facilitate the tensor computation and exchange, which is proven to particularly minimize the excessive communication overhead.

**Strengths:**

1. The idea of redesign the data flow and sharded model training under restrictive computational resources is interesting.
2. The paper has good coherence, and is well-structured.
3. The introduction section explains the concept in an easy and understandable way.
4. The evaluation demonstrates a significant improvement in Model FLOPS utilization compared to DeepSpeed.

**Weaknesses:**

1. Many details about the design and implementation of SpeedLoader are missing. For example, how is the structure and management of sub-batches handled (L105)? The authors claim that "the method allows the computation to be equivalent to gradient accumulation across several sub-batches," but no theoretical proof is provided.
2. The discussion of the hyperparameter tuning strategy in Section 3.3 is unclear. How does the one-shot hyperparameter tuning work? What benefits does it bring? The paper needs to provide a much clearer explanation, as many important implementation details are currently missing.
3. The novelty is marginal, given that much prior art exists on optimization under restrictive computational resources. The paper would benefit from a comparison with more recent methods, such as PagedAttention, to clearly demonstrate differences in performance, resource requirements, and other relevant metrics.
4. Offloading is identified as the most significant factor in addressing the HBM limitations. However, the experimental results show that after 16 GPUs, the trend stabilizes, which does not convincingly demonstrate an enhancement in scalability.
5. Minor issues:
* Please ensure consistency in the capitalization of "FLOPS”, rather than “flops”.
* In Figure 4 where FP and BP denotes -> denote

**Questions:**

1. How does the one-shot hyperparameter tuning work?
2. It would be more convincible with adding more frameworks for detail comparison, for example, PagedAttention, FlashAttention, OLLA.
3. The evaluation is conducted over 16 GPUs, with results appearing to be steady and not demonstrating increased scalability. Could you please include additional GPUs for a more comprehensive evaluation?

**Limitations:**

The authors have adequately addressed the limitation of the potential negative societal impact of their work.
No discussion on limitations, I suggest the authors to add one.

---

> ### Author Rebuttal · Authors · 2024-08-07
>
> ### Response to Question 1
> Thank you for your question. We have prepared a detailed explanation of the one-shot hyperparameter tuning process below. Please review it and let us know if you have any questions, so we can integrate it into the camera-ready version if the paper is accepted.
>
> The primary objective of the one-shot hyperparameter tuning process is to maximize high-bandwidth memory (HBM) and host memory utilization. Given a target context length, the tuner runs two iterations of the training script with different batch sizes, recording the maximum allocated host and device memory during these runs. Since memory consumption is highly linear with respect to the number of input tokens (as discussed in Section 5.1), the tuner calculates the memory increment per input entry. Using this information, it determines the maximum sub-batch size and the number of sub-batches that can be accommodated within the available device and host memory. This approach allows for efficient utilization of memory resources, ensuring optimal performance.
> ### Response to Question 2
> Thank you for your feedback and insightful advice. While SpeedLoader addresses optimization under restrictive computational resources, it also has broader applications. Even with ample hardware resources, the model states for training can exceed the capacity of a single device, making model parallelism essential in most training scenarios.
>
> Recent advances like PagedAttention are primarily focused on optimizing inference rather than training. As of now, ZeRO and FSDP are still the leading paradigms for large language model training involving model offloading and sharding, which highlights SpeedLoader’s solid advantages in these contexts. For reference, we also conducted a trial with Paged Attention and offloading (Fig. S4). However, vLLM is not specifically optimized for offloaded inference, leading to a lower throughput than DeepSpeed.
>
> Additionally, SpeedLoader is compatible with other recent innovations like Flash Attention. Following your suggestion, we integrated Flash Attention into our framework and observed a performance improvement of approximately 5%. We have conducted an ablation study testing the peak performances with this enhancement (Fig. S5). The updated results will be incorporated into the manuscript if the paper is accepted. Experiment details can be found in Author Rebuttal.
> ### Response to Question 3 and Clarification on Weakness 4
> We apologize for any confusion and thank you for your suggestion. While offloading is a significant factor in addressing HBM limitations, SpeedLoader also addresses challenges in sharded model training, such as those encountered in ZeRO and FSDP. Therefore, distribution is another major factor in alleviating HBM pressures.
>
> In Figure 8, we present the **per-device** throughput rather than the overall throughput. This means that while the per-device throughput trend stabilizes after 16 GPUs, the overall training throughput continues to scale linearly with the number of GPUs. For example, training with 32 GPUs can achieve double the throughput compared to a 16-GPU setup. To prevent further confusion, we highlighted per-device nature of data on the y axis label in revised version (Fig. S6).
>
> Therefore, steady **per-device** throughput does not imply a lack of scalability. This phenomenon occurs because more GPUs are physically distant from each other, leading to communication over lower bandwidth interconnections like Ethernet. Given these circumstances, maintaining linear scalability is actually a positive outcome.
>
> We understand that our evaluation with up to 64 GPUs might seem limited compared to some large-scale training setups. Unfortunately, as a research institute, access to larger-scale resources is not always readily available or affordable. We will continue to explore opportunities for testing on a larger scale in the future.
>
> Please let us know if there are any further concerns or if additional clarifications are needed.
> ### Clarification on Weakness 1
> Thank you for your valuable feedback. Below is a revised explanation of the design and implementation details for sub-batching and gradient accumulation in SpeedLoader:
>
> Here we propose SpeedLoader, which processes multiple sub-batches, collectively referred to as an effective batch, with only two model loading and gradient synchronization (Fig. 1, right). **For any incoming batches shaped ($N,l$), they are evenly split to sub-batches along batch size axis ($n_{sb},n,l$), where $N,n_{sb},n,$ and $l$ are effective batch size, number of sub-batches, sub-batch size and sequence length, respectively.**  …
>
> …
>
> To address this issue, **SpeedLoader redesigned the computation graph.** We re-route … switching activations. **In this way, each module computes and accumulates the gradient with respect to input and loss of every individual sub-batch.** This approach enables the automatic differentiation mechanism to function correctly without causing excessive I/O. **This method achieves gradient accumulation with an altered computation graph**, which has significantly reduced …
>
> We will integrate these details into the camera-ready version of the paper upon acceptance. If there are any further clarifications needed, please feel free to ask during the discussion.
> ### Comments on minor issues
> Thank you for pointing out the minor issues. Regarding the confusion between "flops" and "FLOPs," we apologize for not clarifying the difference. The lowercase "flops" refers to "floating point operations," while the capitalized "FLOPs" refers to "floating point operations per second." We will include an explanation of these abbreviations, along with other necessary fixes, in the camera-ready version if the paper is accepted.

---

> ### Author Response · Authors · 2024-08-12
> **TL;DR Summary of Rebuttal**
>
> Dear Reviewer,
>
> Thank you for your thorough review of our manuscript. We appreciate the opportunity to clarify and address your concerns. Below is a summary of the key points in our rebuttal:
>
> - We have revised the manuscript to clarify the tuning and sub-batching processes.
> - Our method achieves the same computational outcomes as gradient accumulation, but with a different order of operations.
> - The primary innovation of SpeedLoader lies in its enhanced efficiency for **distributed and offloaded training** scenarios. This optimization is not limited to offloading; both ZeRO and FSDP without offload can also benefit significantly.
> - While PagedAttention was tested, it was not optimized for offloading, resulting in inference throughput lower than DeepSpeed. Additionally, it is not suitable for training.
> - There may have been some miscommunication regarding Figure 8. The figure depicts **per-device** throughput, not overall throughput. As a result, you can still observe **doubled throughput** on 32 GPUs compared to 16.
> - Additional experiments have been added to examine compatibility and further enhance performance with SOTA techniques. Please refer to the Author Rebuttal for details.
>
> Thank you for your time and effort in reviewing our paper. We look forward to your feedback and hope for a productive discussion.
>
> Best regards

---

> ### Author Response · Authors · 2024-08-14
>
> Dear Reviewer,
>
> Thank you for your time and effort in reviewing manuscripts for NeurIPS this year. We greatly appreciate your contributions.
>
> I am writing to respectfully request a reassessment of our manuscript in light of the clarifications, additions, and results provided in our rebuttal. We believe that our response, along with the global “Author Rebuttal,” addresses the concerns you raised and better clarified some points that might have cause misunderstanding, (e.g., **per-device throughput**).
>
> Once again, thank you for your invaluable contributions to making NeurIPS 2024 a success.
>
> Best regards

---

### Official Review · Reviewer_4D75 · 2024-07-12

**Soundness:** 3
**Presentation:** 2
**Contribution:** 3
**Rating:** 5
**Confidence:** 3

**Summary:**

This paper proposes SpeedLoader, a system for offloading parameters and activations under restricted resources for distributed training and inference.  It utilizes a tensor exchange manager to minimize the communication overheads.  Emperically, a larger proportion of time is spent on computation rather than communication or data movement compared with previous systems, leading to a higher MFU.

**Strengths:**

1. The method does not require training and is only at inference cost.

2. The system has a super linear scalability with the number of GPUs because of the alleviation for PCIe bandwidth.

**Weaknesses:**

Only parameter sharding is considered, other popular scenario like pipeline parallelism covered in FlexGen, tensor parallelism, are not explored here.

**Questions:**

FlexGen has the zig-zag block schedule, which looks similar to the schedule in Figure 1 right side.  Could the author explain their difference?

**Limitations:**

The limitations are acknowledged but not addressed.

---

> ### Author Rebuttal · Authors · 2024-08-07
>
> Response to Question
>
> Thank you for your valuable feedback. While the diagrams may appear similar, SpeedLoader is fundamentally different from FlexGen. FlexGen is a significant work that substantially improves the inference throughput of LLMs by efficiently managing activation and cache tensors. In contrast, our work focuses on the training scenario, which requires carefully designed handling of gradients and the computation graph.
> For simplicity, Fig. 1 only illustrates the forward propagation computation to highlight the throughput advantages. The detailed mechanism of SpeedLoader, including its differences from FlexGen, is demonstrated and explained in Fig. 2.
>
> Additional Comments
>
> Thank you for your insightful comment on other parallelisms. Indeed, both tensor and pipeline parallelism (TP and PP) are prevalent model parallelisms. However, different parallelisms come with distinct challenges. For instance, despite its minimal communication overhead, PP often suffers from bubbles in the computation schedule. On the other hand, tensor parallelism involves heavy peer communication to exchange results. Like FlexGen, SpeedLoader is inherently compatible with these parallelisms. But the primary goal of SpeedLoader is to reduce the heavy burden associated with sharded and offloaded model training, rather than to optimize or alleviate the particular issues of TP and PP.

---

### Official Review · Reviewer_m7wP · 2024-07-16

**Soundness:** 3
**Presentation:** 2
**Contribution:** 3
**Rating:** 6
**Confidence:** 3

**Summary:**

This paper proposes a new paradigm called SpeedLoader for large language models (LLMs) inference and training to offload and reload the layer weights and activations to and from CPU memory. Compared to prior works, SpeedLoader can process multiple batches with the current active layer, which thus able to reduce communication traffic in CPU-GPU transfer and distributed settings. To implement this, SpeedLoader comes with a tensor exchange manager that minimizes memory fragmentation and redundancy. The experimental results show that SpeedLoader can outperform all existing solutions.

**Strengths:**

* SpeedLoader finds a novel way to offload layer weights and activations to CPU memory;
* SpeedLoader achieves the state-of-the-art results compared to existing solutions;

**Weaknesses:**

* No comparison with ideal scenario (i.e. without offloading).
* Experimental details are missing from the paper.

**Questions:**

Thanks for submitting this nice work to NeurIPS, which focuses to solve an important and timely problem. In general, I liked the idea of putting multiple batch together. However, there are a few concerns and comments about the paper:

First, in the experimental results regarding to the performance numbers, there are no comparison with the ideal cases, i.e. without offloading. This comparison is important to understand the positioning of the SpeedLoader, e.g. whether SpeedLoader still have room for optimizations.

Second, in the evaluation plots, the number of tokens in each experiment have not been clearly discussed. This discussion only happens for the functionality test and scalability test. However, showing the (practical) number of tokens in the performance plots are also important because as the number of tokens increases, it might be the case that kernel time dominates (as attention time complexity is quadratic to the number of tokens), which could make SpeedLoader less motivated.

Third, it would be appreciated if a limitation section is presented. This not only helps the reader better positioning the paper, but also inspires future works that improves upon the SpeedLoader.

**Limitations:**

* This work have only evaluations on A100. It is thus unknown how it performs with other generation of GPUs, especially with other PCIe generations that could affect the trade-offs.

---

> ### Author Rebuttal · Authors · 2024-08-07
>
> ### Response to Question 1
>
> Thank you very much for the kind and insightful review. Currently, 7B models at bfloat16 precision is very challenging to be trained on a 40GB A100. The model and optimizer states alone can take up to 56GB memory. One of the methods to deal with it is to shard the model and optimizer across multiple GPUs, as we demonstrated in Fig. 5d,e and Fig. 6 right panel.
>
> Following the advice, we also tested the training MFU of LLaMA-2-7B without sharding and offloading in an 80GB A100 GPU, which is 60.86% MFU under a batch size of 8. We have found that our proposed method (54.59% on the same machine) could achieve a similar MFU as the ideal scenario.
>
> ### Response to Question 2
>
> Sorry for the confusion and thank you very much for this valuable feedback. In our experiments, we compared the performance of SpeedLoader and DeepSpeed under the same maximum token number, which is constrained by the capacity of HBM. This ensures that our benchmarks reflect the best performance for each method. Specifically, we tested 128 entries with a sequence length of 256 per sub-batch (totaling 32k tokens) for SpeedLoader, and 32 entries with a sequence length of 256 per batch (totaling 8k tokens) for DeepSpeed. Given the same HBM size and context length, our memory-efficient approach allows for more effective computation. Given any sequence length, SpeedLoader can support a larger on-device batch size (sub-batch size) than original DeepSpeed, enabling more efficient kernel execution.
>
> To address concerns about the impact of sequence length, we conducted additional tests varying the sequence length on a single A100 40GB with offloading using the LLaMA-2-7B model (Fig. S2 in Author Rebuttal). The results showed that increasing sequence length does not significantly impact both SpeedLoader and DeepSpeed performance, with a variance of no more than 5%. This variation is marginal compared to the 3.29x speedup achieved by SpeedLoader, demonstrating its efficiency and effectiveness.
>
> ### Response to Question 3
>
> Thank you for your suggestion. We highly appreciate your suggestion on the importance of clearly outlining the limitations. We add on a Limitation section with the contents as below and this section will be integrated into the camera-ready version of the paper if it is accepted.
>
> **Limitation**
>
> SpeedLoader provides distributed deep learning settings with an I/O efficient option. By fully utilizing the available host memory, SpeedLoader can significantly reduce the redundant model state exchange among the device, peer and host memory. However, this approach may make host memory a new bottleneck for efficient model operation. Additionally, the implementation of memory allocation in PyTorch only allows sizes that are powers of two, which can lead to inefficient use of memory. For example, if a tensor requires 33 GB, PyTorch allocates 64 GB, potentially leading to out-of-memory issues. Currently, there is no explicit option to modify this behaviour for CPU memory allocation.
>
> Furthermore, while SpeedLoader is theoretically compatible with other parallelism strategies like tensor parallelism and pipeline parallelism, it has not yet been tested in conjunction with these methods. The potential benefits of SpeedLoader in larger-scale hybrid parallel settings remain to be explored. While data parallelisms like FSDP and ZeRO have low communication overhead in exchanging activations, tensor parallelism has lower communication overhead in exchanging parameters. Thus, exploring the dynamic trade-off with speedloader between these parallelisms is a valuable area for future research.
>
> ## Additional Comments
>
> Following your advice on compatibility across generations, to explore the compatibility of SpeedLoader on other platforms, we examined single-device speedups of SpeedLoader on NVIDIA V100S, RTX A6000, and H100 with PCIe Gen 3 to Gen 5 respectively (Fig. S3). The results are 1.37x, 3.56x, and 3.01x, respectively. This proves that our proposed method can have positive effects on different platforms with mixed I/O capability.

---

> > ### Comment · Reviewer_m7wP · 2024-08-09
> >
> > Thanks for the clarification.

---

### Official Review · Reviewer_bDTF · 2024-07-16

**Soundness:** 3
**Presentation:** 3
**Contribution:** 2
**Rating:** 5
**Confidence:** 3

**Summary:**

This paper introduces an innovative compute strategy that minimizes I/O between peers and devices in heterogeneous LLM training, and a high-efficiency tensor manager that optimizes transfers between device and host, reducing fragmentation and redundancy. These enhancements lead to superior inference efficiency compared to existing state-of-the-art methods.

**Strengths:**

1. The topic of this paper is current and timely.
2. The evaluation makes sense. This paper pretrains a 7B and a 13B model following the corresponding configuration of LLaMA-2 and shows better performance than DeepSpeed.

**Weaknesses:**

1. The optimization technique seems to be incremental.

**Questions:**

1. The authors claim SpeedLoader can process multiple batches for one full model loading. What’s the difference from using a larger batch size?
2. What is your experiment configuration of Figure 5? It seems that SpeedLoader minimizes the ratio of Memcpy, yet the ratio of AllGather does not change too much. Do you consider the overlap between Memcpy and Kernel? Why the communication only have a ratio of about 10% in the training scene?

**Limitations:**

Yes, the authors have adequately addressed the limitations.

---

> ### Author Rebuttal · Authors · 2024-08-07
>
> ### Response to Question 1
>
> Thank you very much for the thoughtful insights. The key concept behind SpeedLoader is to achieve **equivalent computation of a large batch** with a small memory consumption. An important feature of SpeedLoader is its management of activations. For instance, processing a batch of 256 sequences, each with a length of 2k, can result in activation data occupying **4GB** between every two layers in the LLaMA-2-7B model, not including the additional space needed for tensors saved for backpropagation. SpeedLoader mitigates this issue by breaking down the workload into sub-batches of 16 samples each, ensuring that only **256MB** of HBM is in active use at any given time, while unused tensors are temporarily moved out of the GPU. Therefore, the computation can be done in **a constant small HBM consumption**.
>
> ### Response to Question 2
>
> We are sorry for the confusion and thanks for pointing it out. The captions might have been not informative in this submission. We have modified it in revised version as below:
>
> **Figure 5** Profiling Results. Panels (**a-c**) were conducted on a single device with offloading; panels (**d-e**) were distributed without offloading. (**a**) DeepSpeed (w/ offloading); (**b**) SpeedLoader (unsaturated-computing, w/offloading); (**c**) SpeedLoader (saturated-computing, w/ offloading); (**d**) DeepSpeed (w/o offloading); (**e**) SpeedLoader (w/o offloading)
>
> The input for unsaturated case has a sub-batch size of 32 and effective batch size of 128; Saturated case has 256, and 2048 for sub-batch size and effective batch size, respectively. Each sample in the batches has a context length of 256.
>
> We implemented meticulous overlapping schedules for communication and computation. Our profiling confirmed that these schedules performed as expected (see Fig. S1 in Author Rebuttal). Figure 5 does not reflect the overlapping schedules, as the pie chart shows the function calling wall time.
>
> The left panels (**a, b, c**) and the right panels (**d, e**) were profiled under different settings. The memcpy in the left panels corresponds to the all_gather and reduce_scatter operations in the right panels as part of the communication payload. The 10.4% all_gather accounts for only a portion of the total communication in sharded training, with an additional 23.5% spent on reduce_scatter for gradients during backpropagation, contributing to the overall communication operations.

---

> ### Author Response · Authors · 2024-08-12
> **TL;DR Summary of Rebuttal**
>
> Dear Reviewer,
>
> Thank you for your hard work in reviewing the manuscript and for your valuable insights. We hope that our rebuttal addresses your concerns. Below is a brief summary:
>
> - SpeedLoader can compute over a large batch with **significantly lower memory requirements**.
> - All_gather is not the only component of communication; reduce_scatter also accounts for a significant portion. Memcpy (in offloaded training) corresponds to the two operations (in non-offloaded training) together. SpeedLoader significantly reduces the reduce_scatter ratio in total wall time.
> - Sequence length has a marginal impact on MFU. Regardless of sequence length, SpeedLoader enables a larger GPU batch size, allowing for more efficient computation.
> - Additional experiments have been added to examine compatibility and further enhance performance with SOTA techniques. Please refer to the Author Rebuttal for details.
>
> Thank you for your time and effort in reviewing our paper. We look forward to your feedback and hope for a productive discussion.
>
> Best regards

---

> ### Author Response · Authors · 2024-08-14
> **Request for Reassessment**
>
> Dear Reviewer,
>
> We sincerely appreciate your valuable contributions to NeurIPS this year and thank you for your time and insights.
>
> We kindly request a re-evaluation of our work based on the detailed clarifications and results provided in our rebuttal. We have addressed your concerns with additional experiments and texts, and hope these clarifications will resolve any misunderstandings, particularly regarding **communication breakdowns**.
>
> Once again, we extend our gratitude for your hard work in making NeurIPS a success this year.
>
> Best regards,

---

### Author Rebuttal · Authors · 2024-08-07

Dear Reviewers,

Thank you for your detailed and insightful feedback. We have conducted additional experiments to enhance the evaluation of SpeedLoader, making it more credible and comprehensive. Below, we address your concerns and clarify some points:

1. **Impact of Different Sequence Lengths Under Full HBM Utilization:** To examine the MFU fluctuation with varying context lengths, we conducted trials with a constant number of total tokens, altering the ratio between batch size and sequence length. The results (Fig. S2) indicate that increasing context length results in a marginal MFU increment (no more than 5% of maximum throughput). This does not diminish SpeedLoader’s advantages, as it consistently allows larger batch sizes, leading to more efficient kernel execution. This observation aligns with our manuscript.

2. **Compatibility Test on Devices with Different I/O Capabilities:** To explore SpeedLoader’s performance on devices with various I/O capabilities and architectures, we tested SpeedLoader on multiple distinct platforms. Due to insufficient collection of GPU options on previous IaaS provider, we sourced the following platforms from different providers:

    | GPU           | PCIe Specification | Source       |
    |---------------|--------------------|--------------|
    | NVIDIA V100S  | Gen 3              | IaaS Provider|
    | NVIDIA RTX A6000 | Gen 4           | Personal     |
    | NVIDIA H100   | Gen 5              | IaaS Provider|

    The results showed 1.37x, 3.56x, and 3.01x speedups on V100S, A6000, and H100, respectively (Fig. S3). The positive speedups showcase robust compatibility across platforms with mixed architectures and capabilities. Note that the MFU being lower on H100 than A100 is expected behavior. As reported by Dao et al., without architecture-specific optimizations like FlashAttention-3, transformers struggle to unleash the full potential of Hopper GPUs. However, the FlashAttention-3 codes are not yet available.

3. **PagedAttention with Offloading:** To provide a more recent reference, we tested the vLLM’s (with PagedAttention) inference throughput with SpeedLoader (Fig. S4). Despite vLLM implementing naïve offloading strategies, it is not highly optimized for this feature. It achieved 22.0 and 4.49 tokens/s throughput for OPT-6.7B and OPT-30B, respectively, and encountered out-of-memory issues when attempting the OPT-175B trial.

4. **Enhanced Performance with FlashAttention-2:** We explored the synergy between SpeedLoader and FlashAttention-2 (Dao et al., 2023). We conducted experiments under identical tuned hyperparameters with FlashAttention-2 enabled and disabled, ensuring maximum hardware utilization. Results (Fig. S5) showed that in distributed offloading scenarios, FlashAttention-2 provides 1.13x, 1.13x, and 1.05x speedups for LLaMA-7B, -13B, and -70B, respectively. In distributed training without offloading, the speedups are 1.16x and 1.13x for 7B and 13B models. For these tests, we used 1, 2, and 4 nodes with 16 NVIDIA A100-40GB GPUs each for 7B, 13B, and 70B trials, respectively. With this optimization and the upcoming FlashAttention-3, SpeedLoader can achieve throughput very close to non-sharded and non-offloaded training.

To prevent misunderstanding and confusion, we made some adjustments and additions to the text (details are presented in individual responses):

1. **Revised Section 3:** We have provided more information on the design of SpeedLoader to help readers better understand its mechanism.

2. **Added Limitation Section:** To help readers better position the paper and inspire future work, we have added a limitation section:

   **Limitation:** SpeedLoader provides distributed deep learning settings with an I/O efficient option. By fully utilizing the available host memory, SpeedLoader can significantly reduce the redundant model state exchange among the device, peer and host memory. However, this approach may make host memory a new bottleneck for efficient model operation. Additionally, the implementation of memory allocation in PyTorch only allows sizes that are powers of two, which can lead to inefficient use of memory. For example, if a tensor requires 33 GB, PyTorch allocates 64 GB, potentially leading to out-of-memory issues. Currently, there is no explicit option to modify this behaviour for CPU memory allocation.

   Furthermore, while SpeedLoader is theoretically compatible with other parallelism strategies like tensor parallelism and pipeline parallelism, it has not yet been tested in conjunction with these methods. The potential benefits of SpeedLoader in larger-scale hybrid parallel settings remain to be explored. While data parallelisms like FSDP and ZeRO have low communication overhead in exchanging activations, tensor parallelism has lower communication overhead in exchanging parameters. Thus, exploring the dynamic trade-off with SpeedLoader between these parallelisms is a valuable area for future research.


3. **Abbreviation Explanation:** We have added an explanation to help readers differentiate between floating-point operations (flops) and floating-point operations per second (FLOPs).

4. **Grammatical Errors:** We have corrected various grammatical errors throughout the manuscript.

The above changes and additional results will be integrated into the camera-ready version of the manuscript if the paper is accepted. Thank you again for reviewing this paper and for the valuable feedback provided.

Best Regards

---

### Decision · Program_Chairs · 2024-09-25

**Decision:**

Accept (poster)

**Comment:**

Reviewers agree this paper addresses an important and timely problem.

Reviewers initial criticisms (e.g. lack of comparison on other platforms, lack of discussion about FlexGen and PagedAttention) are well addressed in the rebuttal, and we expect to be well addressed in a final version.